# Assessing feasibility and maternal acceptability of a biomechanically-optimized supine birth position: A pilot study

Lisa Bouille[1,☯], Joanna Sichitiu[1,☯], Julien Favre[2], David Desseauve[1]*

1 Women-Mother-Child Department, Lausanne University Hospital, Lausanne, Vaud, Switzerland,
2 Department of Musculoskeletal Medicine, Swiss BioMotion Lab, Lausanne University Hospital and University of Lausanne (CHUV-UNIL), Lausanne, Vaud, Switzerland

☯ These authors contributed equally to this work.
* david.desseauve@chuv.ch

**Data Availability Statement:** The data underlying the findings in this study cannot be made freely available due to ethical and legal restrictions. An important number of variables included in this

## Abstract

### Background

In order to manage a protracted second stage of labor, "eminence-based" birth positions have been suggested by some healthcare professionals. Recent biomechanical studies have promoted the use of an optimized supine birthing position in this setting. However, uncertainty exists regarding the feasibility of this posture, and its acceptability by women. This pilot study primarily aimed to assess these characteristics.

### Objective and methods

In this monocentric prospective study, 20 women with a protracted second stage of labor were asked to maintain a biomechanically-optimized position for at least 20 minutes at full dilatation. This posture is similar to the McRoberts' maneuver. Maintaining the position for 20 minutes or more was considered clinically relevant and indicative of feasibility and acceptability. Satisfaction with the position was assessed using a Visual Analogue Scale (VAS). A sub-group analysis was performed to assess eventual differences between more and less satisfied patients, according to the median of patients' satisfaction scores.

### Results

Seventeen patients (85%) maintained the optimized position for at least 20 minutes. The median satisfaction score of these participants was 8 (interquartile range: 1) out of 10. No significant differences were found between the two sub-groups (satisfaction score <8 vs satisfaction score ≥8) regarding general and obstetric characteristics, as well as obstetrical and fetal outcomes.

### Conclusion

The optimized position is acceptable and feasible for women experiencing a protracted second stage of labor. Further clinical studies are needed to assess the efficiency of such positions when women undergo an obstructed labor.

study could be used to identify participants. Therefore, the Swiss Association of Research Ethics Committee strictly forbids making such data freely available. A request for data could be made to the data science département of CHUV (dsr. data@chuv.ch) after acceptance by our local ethical commission (scientifique.cer@vd.ch).

**Funding:** Leenaards fondation was associated directly with this study.

**Competing interests:** The authors report no conflicts of interest.

## Introduction

The increased rate of cesarean section in developed countries is a worrying public health problem. According to the World Health Organization, there is no reduction in maternal morbidity and mortality at rates greater than 10–15%, a threshold largely exceeded in developed countries [1]. Nearly 20% of cesarean sections are performed in response to an obstructed labor after a protracted second stage of labor, corresponding to when the fetus does not engage in the pelvis despite a fully dilated cervix [2]. This highlights the need for further research on childbirth biomechanics, which could elucidate the mechanism behind such obstructions and point to more favorable birthing positions.

In order to manage protracted labor, some healthcare professionals have suggested adopting "eminence-based" positions for giving birth. Eminence-based medicine, as opposed to evidence-based medicine, refers to clinical practices relying entirely on the opinion and experience of health professionals, rather than on scientific research. Recently, there is growing evidence recognizing that eminence-based positions can optimize the alignment between the birth canal (formed by pelvic bones, lumbar spine and soft pelvic tissues) and the fetus [3–5]. Some positions may promote fetal descent through the pelvic inlet (the anatomical limit between the false and true pelvis, bounded anteriorly by the pubic symphysis, laterally by the iliopectineal lines, and posteriorly by the sacral promontory) more efficiently than others [3–5]. In theory, an optimal birth position can be biomechanically defined as a position where the pelvic inlet is perpendicular to the lumbar spine, while minimizing lumbar lordosis i.e. achieving a sort of "obstetric chute" [3–5]. Recent research in experimental settings (i.e. not during labor), using an optoelectronic motion capture system, suggested a specific position which approaches the biomechanical ideal mentioned above [3–5]. This method involves a squatting position in a supine set-up i.e. lying on one's back with a 30 degree angle from horizontal, legs in abduction with maximal hip flexion and minimizing lumbar lordosis (Fig 1). However, this mechanistic approach to childbirth based only on biomechanics is insufficient, as other factors at play during childbirth and the clinical impact of such positions have yet to be assessed [6, 7]. Consequently, there is a need to determine the feasibility of optimized positions in clinical settings to ensure that fundamental biomechanical research can be applied, especially during labor dystocia [6]. The acceptability of biomechanically-optimized positions should also be evaluated as these more complex positions could make a parturient feel restrained in their experience of giving birth.

This prospective pilot interventional study primarily aimed to assess the feasibility and acceptability for parturients of a biomechanically-optimized birth position. We hypothesize that participants can maintain the optimized birth position, resembling the posture obtained at the end of a *McRoberts' maneuver* [6, 8], for at least 20 minutes without feeling uncomfortable. In addition, for the women who maintained the position for at least 20 minutes, we investigated if population characteristics, obstetrical conditions and obstetrical outcomes differed between women reporting higher and lower satisfaction.

## Materials and methods

### Study population

Participants gave birth at a tertiary university hospital (Lausanne University Hospital) between August 2019 and December 2019. Eligible participants were recruited during their antenatal consultation in our outpatient clinic or before induction of labor for non-emergency indications (post-term, oligoamnios, prelabor rupture of membranes) at greater than 39 weeks of gestation. To be eligible, women had to have been diagnosed with a protracted second stage of

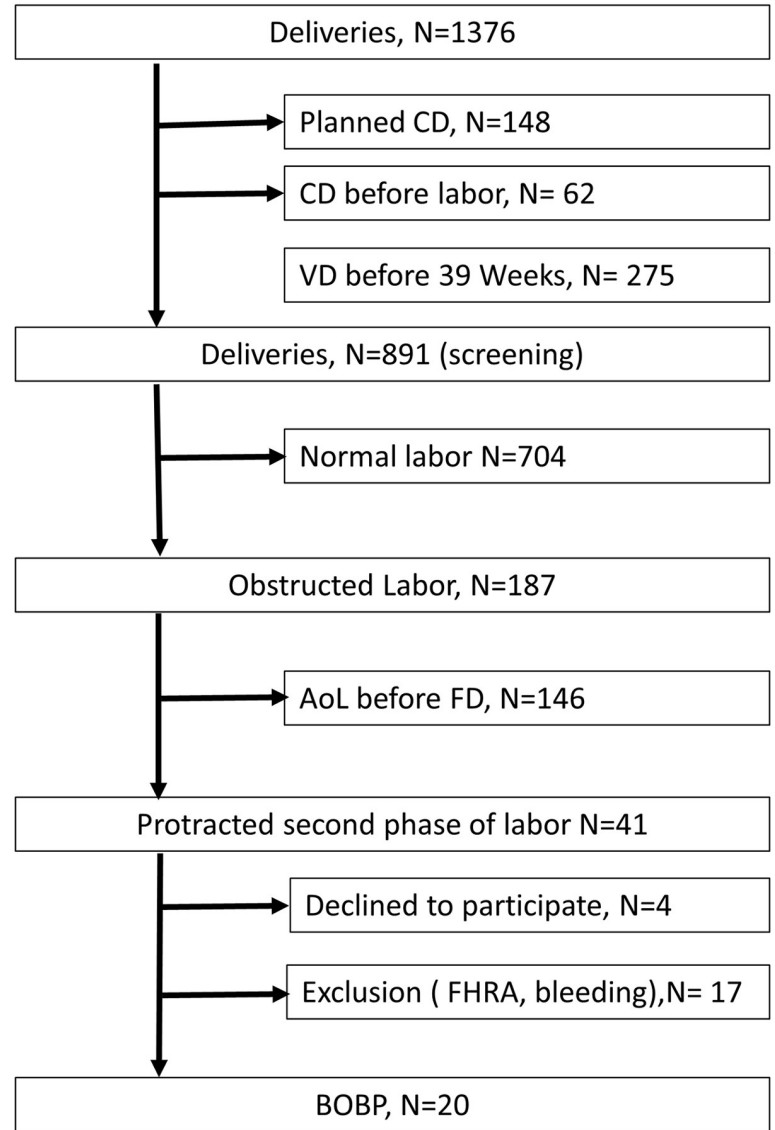

AoL, Arest of Labor; BOBP, Biomechanically-Optimized Birth Position; CD, cesarean delivery; FD, Full Dilatation; FHRA, Fetal Heart rates abnormalities; VD, Vaginal delivery; Weeks, weeks of gestation.

**Fig 1. Illustration of the biomechanically-optimized supine position assessed in this study.**

labor, i.e. those whose fetus did not engage in the pelvic inlet after one hour at full cervical dilation. Exclusion criteria included physical incapacity or medical contraindications precluding the intervention (e.g. orthopedic injury or disease preventing the parturient from adopting the optimized position), clinically significant concomitant diseases, incapacity of judgment, inability to follow the procedures of the study due to language barriers/psychological disorders and non-reassuring fetal heart monitoring or abnormal bleeding during labor.

The study protocol was approved by the local Institutional Review Board (Ethical Commission of the Canton of Vaud, Switzerland, 2019–00872) and declared on *ClinicalTrials.gov*

portal: NCT04056793 (14/08/2019). All experiments were performed in accordance with relevant Ethics Committee and relevant Swiss guidelines and regulations. All participants provided written informed consent. Also, the individual in Fig 1 gave written informed consent (as outlined in PLOS consent form) to publish this photograph. No financial incentive was offered for participation. A physician independent of the research team performed data monitoring and quality assurance.

## Intervention

When a protracted second stage of labor occurred, the parturient was asked to adopt the bio-mechanically-optimized birth position with the help of a midwife. The optimized position performed in this study was previously defined and biomechanically assessed by our team [3]. This supine position combined maximal flexion and abduction of the hip joints with minimizing lumbar lordosis, using foot rests to correctly position the legs (Fig 1) [3]. Participants were invited to maintain this position for 20 minutes. This period of time was selected because it represents approximately 10 uterine contractions and we considered that such a duration would allow us to appreciate the effectiveness of the intervention without being too constraining for participants. This duration was also selected because two-thirds of women in our hospital request epidural analgesia and it is not recommended to make them adopt such a position for a prolonged amount of time, in order to avoid any risk of malposition or nerve compression [9]. Women were free to change position at any time, or they could maintain it for more than 20 minutes if they so wished. The study did not change the usual management of dystocia, in particular the administration of oxytocin after 1 hour without progression of the presentation at the second stage of labor (protracted second stage of labor).

## Outcomes

The primary outcome was to assess the feasibility of the optimized position, evaluated according to the duration the participants could maintain it. Specifically, the position was considered feasible for a participant when the participant maintained it for at least 20 minutes. The secondary objective was to assess the acceptability (subject satisfaction and pain perceived) of the optimized position.

We used the VAS (Visual Analogue Scale) to evaluate participants' satisfaction regarding the position, ranging from 0 (no satisfaction) to 10 (complete satisfaction). An interview was conducted by one of the authors (LB) during the two days following delivery, where a standardized question was asked of all the participants: *"Could you report your satisfaction about the optimized position that was proposed to you on this visual scale"*. The occurrence of pain related to the position was also recorded. A two-day time frame was chosen as it allowed patients to accurately recall their satisfaction concerning the optimized position without the study being too intrusive.

A sub-group analysis was performed to compare population characteristics, obstetrical conditions and obstetrical outcomes between the women with higher satisfaction scores and those with lower satisfaction scores. To this end, the participants who could maintain the posture for at least 20 minutes were divided in two sub-groups using the median satisfaction score as a cut-off. Data concerning risk factors of non-engagement of the fetus at full dilation, delivery outcome, fetal outcome and occurrence of eventual adverse events, such as symphysis pubis diastasis or maternal peripheral nerve disorders, were also collected. Perineal tears are reported according to the Royal College of Obstetrician and Gynecologist (RCOG) classification [10].

## Statistical analysis

Power calculations were performed. We hypothesized that future studies involving newly developed technology focusing on the optimization of birth posture should be acceptable and feasible if 80% of participants could maintain a specific position for 20 minutes while giving birth. We estimated that 20 women would provide a power of at least 80% to detect a relative difference of 75% or greater in the incidence of the primary outcome (i.e. 80% of women maintaining the optimized position and 20% who did not tolerate it), with a 5% two-sided type I error. According to recent data, protracted labor at full dilatation affects 2.7% of women [11]. We estimated that 5 months would been necessary to complete this study in our maternity unit (average of 250 births per month).

Statistical distribution of quantitative variables was assessed with a Shapiro-Wilk test. In view of the non-normal distribution of the data, results are reported as medians and interquartile ranges (IQR) for numeric variables or as numbers and percentages for categorical variables. For quantitative variables, the statistical significance of the differences between the subgroups was tested using Wilcoxon rank-sum tests. For qualitative variables, p-values were calculated using a multilevel mixed-effects linear regression. The significance level was defined as $p < 0.05$. Data analysis and reporting were performed using Stata V16 (Stata Corp, College Station, TX, USA).

## Results

During the study period, 891 women were screened, out of which 20 were included (2.2%) (Fig 2). The median age of the participants was 33 (IQR: 7.5) years, and the median gestational age at labor was 40 (IQR: 2) weeks. Ninety percent of patients were nulliparous. Of the 20 patients included in the study, 13 (65%) had fetuses in the left and 7 (35%) in the right occiput anterior position. All women had epidural analgesia at some point before testing the biomechanically-optimized posture and 4 of them additionally had spinal anesthesia due to inadequate pain relief (Table 1).

Seventeen women (85%) stayed in the optimized position for a minimum of 20 minutes. One woman could not maintain the position for more than 5 minutes due to pain related to fetal engagement. A second woman discontinued after 10 minutes due to non-reassuring fetal heart rate. A third participant could not maintain the position for more than a few seconds due to low back pain related to labor that increased in any supine position. Specific pain/discomfort associated with the optimized position occurred in 2 participants (11.8%).

Among the 17 participants who maintained the optimized position for at least 20 minutes, the median satisfaction score was 8 (IQR: 1) out of 10. Neither maternal peripheral nerve dysfunction nor symphysis pubis diastasis were reported in any of these patients. Regarding obstetrical outcomes, 14 women achieved a spontaneous vaginal delivery (82.3%), 2 women underwent cesarean section (11.7%) and 1 woman required assisted delivery using forceps for non-progression of the presentation (5.9%). At the time of birth, the median Apgar score of the babies at 1, 5 and 10 minutes was 8.5, 9.5 and 10 (IQR: 2–1.5–0.5) respectively. Their median weight and head circumference were respectively 3610 (IQR: 520) g and 35 (IQR: 1.5) cm (Table 2).

Comparisons of population characteristics, obstetrical conditions and obstetrical outcomes between the women with higher and lower satisfaction scores regarding the optimized position (median satisfaction score of VAS = 8 used as cut-off) are reported in Table 3.

No significant differences were found between the two sub-groups regarding general and obstetric characteristics, obstetrical and fetal outcomes or occurrence of specific pain related to the optimized position ($p \geq 0.08$).

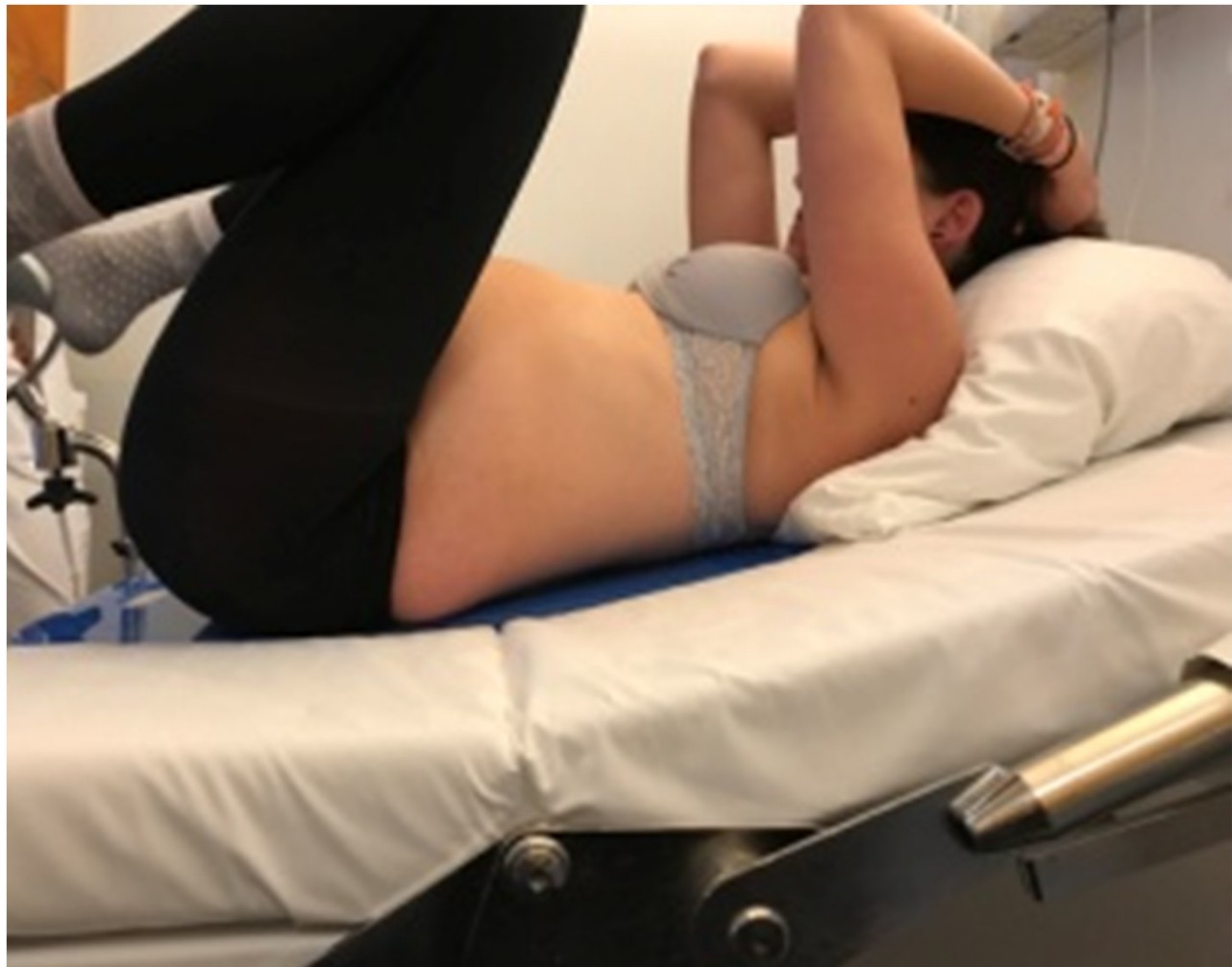

**Fig 2. Flow chart.**

Midwives who positioned participants did not report any issue concerning the study's completion, being already experienced in advising various positions when a mechanical dystocia occurs and in helping patients to adopt them. Therefore, the intervention did not interfere with global patient care.

## Discussion

Most of the participants (85%) could maintain the biomechanically-optimized position for at least 20 minutes, with a good satisfaction score (VAS median score at 8 out of 10). Pain specifically related to this position occurred only in a minority of cases. No dysfunction of peripheral nerves nor symphysis pubis diastasis were reported in any of the patients. Therefore, it seems that this prospective pilot study confirmed the hypothesis that an optimized position resembling squatting while lying supine, similar to that obtained at the end of a *McRoberts' maneuver* [3, 5], is feasible and well accepted by women in labor. Furthermore, there was no difference in participant characteristics between the women reporting higher and lower levels of satisfaction.

**Table 1. Demographic and obstetrical characteristics just before adopting the biomechanically-optimized position.**

| Study population (n = 20) | |
|---|---|
| **Population characteristics** | |
| Maternal age (years) | 33 [7.5] |
| Nulliparous | 18 (90%) |
| Gestational age (weeks) | 40 [2] |
| BMI (kg/m$^2$) | 23.5 [5.2] |
| Maternal height (cm) | 161.5 [10.5] |
| Maternal weight (kg) | 64.5 [16.2] |
| **Obstetrical conditions** | |
| Oxytocin (UI) | 1.5 [2] |
| SFH (cm) | 35.5 [3] |
| Induction of labor | 16 (80%) |
| AROM | 7 (35%) |
| Cephalic presentation | |
| *LOA* | 13 (65%) |
| *ROA* | 7 (35%) |
| Regional analgesia | 20 (100%) |
| Spinal anesthesia | 4 (20%) |
| Fetal macrosomia suspicion | 4 (20%) |

All data shown as n (%) or median [IQR].

*BMI*: *Body mass index*, *SFH*: Symphysis-Fundal Height, *AROM*: Artificial Rupture of Membranes, *LOA*: Left Occipito-Anterior, *ROA*: Right Occipito-Anterior.

A high percentage of nulliparous participants emerged in the population characteristics (90%). This could be explained by a noticeably longer second stage of labor in nulliparous women and by a higher prevalence of mechanical dystocia in this population (16.5% of nullipara versus 9.9% of multipara) [12, 13]. Naturally, engagement of the fetus and childbirth in parous women occurred more frequently before one hour of full cervical dilatation.

Another factor associated with higher risk of obstructed labor is the presence of an occiput posterior position [14]. Interestingly, in our cohort, fetuses in the occiput anterior position at time of birth were over-represented in comparison to the rates at full dilatation reported in the literature [14–17]. Per the protocol in place at our hospital, manual rotation of occiput posterior positions is only performed after one hour of full dilation. Despite the occurrence of obstructed labor, all participants had fetuses in «spontaneous» occiput anterior positions.

According to the Swiss Federal Statistical Office, rates of emergency cesarean section and instrumental deliveries in Switzerland reached 15.8% and 11.1% respectively in 2017 [18]. Therefore, this small sample size study seemed consistent with usual practice: it did not illustrate abnormal rates of cesarean section or assisted delivery. Our study was not designed to assess the impact of this position on the means of delivery, but the results it provided motivate further studies to address this question.

One limitation of this study is that no reference value differentiating between a birth position judged satisfactory or unsatisfactory by the parturient is available in the literature. In light of this limitation, we chose the median of the satisfaction score to compare women with higher and lower satisfaction. This approach nonetheless allows us to draw a parallel between our results and previous randomized trials which reported maternal satisfaction with position during labor in order to correct occiput posterior fetal position [15–17, 19]. This indicates that the

**Table 2.  Outcomes for the 17 women who maintained the optimized position for at least 20 minutes.**

| Study population (n = 17) | |
|---|---|
| **Biomechanically-optimized position outcomes** | |
| Duration (min) | 26.2 [5] |
| Satisfaction score (from 0 to 10) | 8 [1] |
| Reported pain | 2 (11.8%) |
| **Obstetrical outcomes** | |
| Spontaneous delivery | 14 (82.3%) |
| Instrumental delivery | 1 (5.9%) |
| Caesarean section | 2 (11.7%) |
| Perineal tears (RCOG Classification) | |
| 0 | 6 (37.5%) |
| 1 | 1 (6.2%) |
| 2 | 8 (50%) |
| 3 | 1 (6.2%) |
| **Fetal outcomes** | |
| Apgar score (from 0 to 10) at: | |
| 1 minute | 8.5 [2] |
| 5 minutes | 9.5 [1.5] |
| 10 minutes | 10 [0.5] |
| Weight (g) | 3510 [520] |
| Head circumference (cm) | 35 [1] |

All data shown as n (%) or median [IQR].

*RCOG*: Royal College of Obstetricians and Gynaecologists.

acceptability of the biomechanically-optimized position in this study is at least comparable to the results reported in prior works [15–17, 19]. Another limitation of this study concerns the small size of our cohort, which didn't allow us to adjust potential confounding factors for satisfaction and acceptability (parity, duration of labor).

Despite these limitations, our results are already consistent enough to confirm that the biomechanically-optimized position presented in this study is suitable and should be further assessed. Additionally, to improve the assessment of the birthing position, it would be useful to consider other evaluation criterion, such as the progression of the fetus in the pelvic inlet. Furthermore, computational models could also provide crucial insights on the different biomechanical aspects of such positions [20]. Indeed, such models could inform us on the efficacy of the positions and constitute a valid step before conducting large interventional studies [6, 21]. Future studies should also focus on the factors impacting the acceptability and satisfaction of the optimized position. For instance, it would be useful to understand the role of pain relief and its effect on the sense of self-empowerment. Better understanding of the factors related to acceptability and satisfaction is important, to avoid solutions exclusively based on biomechanics and therefore potentially disempowering to women's experience during delivery [22]. Labor ward caregivers must then endeavor to cultivate maternal instincts and advocate non-intervention in normal processes [22]. Nonetheless, whether instinctively adopted or not, it remains important to understand why some postures are more favorable than others.

## Conclusion

The majority of our participants were able to maintain the biomechanically-optimized birthing position for a duration considered to be clinically relevant, without side effects. Therefore, this

**Table 3. Comparison of population characteristics, obstetrical conditions and obstetrical outcomes between women with higher and lower satisfaction scores regarding the optimized position.**

| Study population (n = 17) | Satisfaction score <8 | Satisfaction score ≥ 8 | p-value |
|---|---|---|---|
| | n = 8 | n = 9 | |
| **Population characteristics** | | | |
| Maternal age (years) | 30 [9] | 33 [5] | 0.4 |
| Nulliparous | 8 (100%) | 8 (88.9%) | 0.3 |
| Gestational age (weeks) | 40 [2] | 40 [1] | 0.2 |
| BMI (kg/m$^2$) | 23.6 [8.5] | 23.3 [4.4] | 0.6 |
| Maternal height (cm) | 160 [8] | 163 [8] | 0.7 |
| Maternal weight (kg) | 59 [23.5] | 67 [9] | 0.7 |
| **Obstetrical conditions** | | | |
| Oxytocin (UI) | 1.7 [1.2] | 1.7 [3.8] | 0.8 |
| SFH (cm) | 35 [4] | 36 [2] | 0.7 |
| Induction of labor | 7 (87.5%) | 6 (66.7%) | 0.3 |
| AROM | 2 (25%) | 3 (33.3%) | 0.7 |
| Cephalic presentation | | | |
| *LOA* | 4 (50%) | 6 (66.7%) | 0.5 |
| *ROA* | 4 (50%) | 3 (33.3%) | |
| Epidural analgesia | 8 (100%) | 9 (100%) | 1.0 |
| Spinal anesthesia | 0 (0%) | 3 (33.3%) | 0.7 |
| Fetal macrosomia suspicion | 0(0%) | 3 (33.3%) | 0.7 |
| **Optimized position outcomes** | | | |
| Duration (min) | 20 [2.5] | 25 [10] | 0.3 |
| Reported pain | 2 (25%) | 0 (0%) | 0.08 |
| Satisfaction score (from 0 to 10) | 7 [1] | 8 [1] | n/a |
| **Obstetrical outcomes** | | | |
| Spontaneous delivery | 7 (87.5%) | 7 (77.8%) | 0.8 |
| Instrumental delivery | 0 (0%) | 1 (11.1%) | |
| Cesarean section | 1 (12.5%) | 1 (11.1%) | |
| Perineal tears (RCOG classification) | | | |
| 0 | 2 (28.5%) | 4 (44.4%) | 0.25 |
| 1 | 0 (0%) | 1 (11.1%) | |
| 2 | 4 (57.1%) | 4 (44.4%) | |
| 3 | 1 (14.3%) | 0 (0%) | |
| **Fetal outcomes** | | | |
| Apgar score (from 0 to 10) | | | |
| 1 minute | 8 [4] | 9 [2] | 0.5 |
| 5 minutes | 10 [4] | 9 [1] | 0.8 |
| 10 minutes | 10 [2] | 10 [0] | 0.1 |
| Weight (g) | 3350 [360] | 3700 [380] | 0.18 |
| Head circumference (cm) | 34.5 [1.4] | 35 [1.3] | 0.3 |

All data shown as n (%) or median [IQR].

*BMI*: *Body mass index*, *SFH*: Symphysis-Fundal Height, *AROM*: Artificial Rupture of Membranes, *LOA*: Left Occipito-Anterior, *ROA*: Right Occipito-Anterior.

position can be considered as feasible and acceptable for women giving birth. These findings call for additional biomechanical and interventional studies to determine if this optimized position is an efficient solution to obstructed labor.

## Supporting information

**S1 Checklist.**
(PDF)

**S1 Protocol.**
(PDF)

## Author Contributions

**Conceptualization:** Lisa Bouille, David Desseauve.

**Data curation:** Lisa Bouille.

**Formal analysis:** Lisa Bouille, Joanna Sichitiu, David Desseauve.

**Funding acquisition:** David Desseauve.

**Investigation:** Lisa Bouille.

**Methodology:** Joanna Sichitiu, Julien Favre, David Desseauve.

**Project administration:** David Desseauve.

**Supervision:** Julien Favre, David Desseauve.

**Validation:** David Desseauve.

**Writing – original draft:** Lisa Bouille, David Desseauve.

**Writing – review & editing:** Joanna Sichitiu, Julien Favre, David Desseauve.

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
