## [Decision Letter · Decision Letter 0]

14 Apr 2021

PONE-D-20-31050

Feasibility and acceptability assessment of a biomechanically-optimized supine birth position: A pilot study

PLOS ONE

Dear Dr. Desseauve,

Thank you for submitting your manuscript to PLOS ONE. After careful consideration, we feel that it has merit but does not fully meet PLOS ONE’s publication criteria as it currently stands. Therefore, we invite you to submit a revised version of the manuscript that addresses the points raised during the review process.

The reviewers raised several minor issues, including a request for clarification on some of the statistical methods, the need for some corrections in phrasing, and additional references to be included. The reviewers' comments can be viewed in full, below.

We look forward to receiving your revised manuscript.

Kind regards,

Natasha McDonald, PhD

Associate Editor

PLOS ONE

Journal Requirements:

5. Please include captions for your Supporting Information files at the end of your manuscript, and update any in-text citations to match accordingly. Please see our Supporting Information guidelines for more information: http://journals.plos.org/plosone/s/supporting-information

6. We note that Figure 2 includes an image of a participant in the study. 

As per the PLOS ONE policy (http://journals.plos.org/plosone/s/submission-guidelines#loc-human-subjects-research) on papers that include identifying, or potentially identifying, information, the individual(s) or parent(s)/guardian(s) must be informed of the terms of the PLOS open-access (CC-BY) license and provide specific permission for publication of these details under the terms of this license.

Please download the Consent Form for Publication in a PLOS Journal (http://journals.plos.org/plosone/s/file?id=8ce6/plos-consent-form-english.pdf). The signed consent form should not be submitted with the manuscript, but should be securely filed in the individual's case notes.

Please amend the methods section and ethics statement of the manuscript to explicitly state that the patient/participant has provided consent for publication: “The individual in this manuscript has given written informed consent (as outlined in PLOS consent form) to publish these case details”.

Reviewers' comments:

Reviewer's Responses to Questions

**Comments to the Author**

1. Is the manuscript technically sound, and do the data support the conclusions?

Reviewer #1: Partly

Reviewer #2: Yes

Reviewer #3: Yes

Reviewer #4: Yes

2. Has the statistical analysis been performed appropriately and rigorously? 

Reviewer #1: No

Reviewer #2: Yes

Reviewer #3: Yes

Reviewer #4: Yes

3. Have the authors made all data underlying the findings in their manuscript fully available?

Reviewer #1: No

Reviewer #2: Yes

Reviewer #3: Yes

Reviewer #4: Yes

4. Is the manuscript presented in an intelligible fashion and written in standard English?

Reviewer #1: Yes

Reviewer #2: Yes

Reviewer #3: No

Reviewer #4: Yes

5. Review Comments to the Author

Reviewer #1: This manuscript reports observed results from a pilot study investigating feasibility and acceptability of a biomechanically-optimized supine birth position. I have below questions and comments for statistical analysis.

The description is not clear for power calculation. Please provide detail for the hypotheses. What will you compare to 80%?

For qualitative variable, why was a multilevel mixed-effects linear regression used? If the qualitative variable is categorical, frequency table and Chi-squared test or Fisher’s exact test may be used.

Reviewer #2: Thanks for giving me such a valuable chance to review this paper. This paper addressed an important issue in maternity care and is very well-written. The authors assessed the feasibility and maternal acceptability of a biomechanically-optimized supine birth position. Similar studies on this topic are limited; Thus this study has academic significance. On the other hand, This study also provides valuable insights into dealing with obstructed labor.

On the whole, I only have a few suggestions for revision. Comments are as following. I would be delighted to recommend this paper for publication after a minor revision.

Background:

#1 What are“eminence-based” positions? Any examples?

#2 “ there is growing evidence recognizing that such management can improve the alignement between the birth canal (formed by pelvic bones, lumbar spine and soft pelvis tissues) and the fetus.”

Please consider adding references to support this statement.

#3As is stated in the Background section, “a particular position, similar to squatting while lying on the back” could optimize delivery.

How does this position optimize delivery? In what manner? By improving the alignement between the birth canal? I believe a brief explanation may add values.

Material and method

#4 Who performed the interview two days after delivery? Any rationale for choosing this date to conduct the interview?

“standardized question was ask to all the participants during an interview that took place during the two days following the delivery.

Discussion

#5 It would be better to add a reference for the numbers

“According to the Swiss Federal Statistic Office, rates of emergency cesarean section and instrumental deliveries in Switzerland reached respectively 15.8 % and 11.1 % in 2017”

Reviewer #3: General Notes:

-- The position is not McRobert position -- it is McRoberts' position (or McRoberts' maneuver). It was named for Dr. William McRoberts. It can be written without the possessive apostrophe, if desired -- but the s is required.

-- Information on how this position was maintained (e.g. use of foot rests) would be helpful (referring to Figure). Those who understand McRoberts' position normally expect it to be maintained by two assistants/family members (one on each leg).

-- While you refer to your papers on an optimal birth position, it would be very helpful to provide a little more detail in the Introduction of this paper about the findings in those papers. What is the necessary angle of the pelvis? How did you define optimized? How is it different in angles from lithotomy?

Substantive Corrections or Clarifications:

INTRODUCTION:

-- I am not sure what you mean by "the pelvic inlet plan" -- please clarify

RESULTS:

-- what is "important back discomfort"?

-- it appears from the Results section that when you discussed peripheral nerve disfunction, that you were thinking about maternal nerves. This should be clarified in the Methods section (outcomes paragraph). Readers familiar with the fetal risk factors associated with labor dystocia and the typical use of McRoberts' maneuver in shoulder dystocia events may assume that the description refers to neonatal nerve injuries.

-- I would suggest putting a percent sign in the parenthetical values of Table 1 to remind readers what you are signifying, as it is a bit hard to ascertain when first looking at the table.

-- I would suggest separating out the 3 time points for Apgar scores onto three separate lines in Table 2 and Table 3. This will clarify it for readers.

DISCUSSION:

-- in your paragraph discussing the limitation due to the lack of a literature value on position, I originally thought that you were talking about the McRoberts' position. You should clarify in this sentence that you are discussing the subjects' opinions regarding their satisfaction with the position.

-- You say that you are comparing your results to randomized trials related to effectiveness of various positions (refs 14-17). However, you have not reported on any data regarding effectiveness -- only patient satisfaction. If there are specific data that you are comparing to (e.g. patient satisfaction with the positions used in those studies), please cite that data and clearly describe the comparison.

-- "while further studies are necessary to precise the numbers" is not grammatically correct - and I am not sure exactly what you are trying to say. Increasing the numbers would allow you to statistically differentiate between narrower groups -- but it is not likely to increase the precision of your measurements. Please write this section more clearly to differentiate between actual measurements and statistical conclusions.

Grammatical Corrections (NOTE: There are additional grammar errors that I did not note. Please have the document reviewed by a proficient writer whose first language is English to minimize errors.)

ABSTRACT:

-- it is most common to refer to the second stage as "second stage OF labor" rather than without the preposition

-- "is similar to the McRobert maneuver" should be "is similar to the McRoberts' maneuver"

-- In "A sub-group analyses was performed according to the median of the patient satisfaction score, to assess eventual differences between more and less satisfied patients," analyses should be ANALYSIS and the comma should be deleted between "score" and "to"

INTRODUCTION:

-- "threshold largely exceeded in developed countries" should be preceded by "a" as in "a threshold largely exceded..."

-- change "approach of childbirth" to "approach to childbirth"

-- delete "in future researches." If you really want to keep it (it doesn't make sense), it should be "in future research."

-- "can be transfer to" should be "can be transferred to"

-- "could make parturient feel" should be "could make a parturient feel"

-- "resembling at the posture" should be "resembling the posture"

MATERIALS AND METHODS:

-- "prelabor rupture of membrane" should be "prelabor rupture of membranes"

-- "whose fetus did not engage" should be "those whose fetus ..."

-- in the discussion of the IRB approval, there should be no comma between the IRB protocol number and "and" [i.e. -00872) and declared] and there should be no period at the end of ClinicalTrials.gov

-- "would allow to appreciate the effectiveness of the intervention without being to constraining for the women" should be "would all us to appreciate ... without being too constraining ..."

-- "the primary outcomes" should be "the primary outcome"

-- if you want to include the comma in the first sentence about outcomes, then you need a subject in the second phrase (i.e. "and this was evaluated.")

-- "to evaluate participant's satisfaction" should be either "to evaluate a participant's satisfaction" or "to evaluate participants' satisfaction"

-- "A standardized question was ask to all" should be "A standardized question was asked of all"

-- "obstetrical conditions and obstetrical outcomes between women the women with higher satisfaction to the women with lower satisfaction" should be "obstetrical conditions and obstetrical outcomes between the women with higher satisfaction and the women with lower satisfaction."

-- "Data concerning risks factor of non-engagement" should be "Data concerning risk factors of non-engagement" and "peripheral nerves disorders" should be "peripheral nerve disorders"

-- "Shapiro-wilk test" should be "Shapiro-Wilk test" as it refers to a name (or pair of names)

RESULTS:

-- "median weight and head circumference were respectively of 3610" should be "median weight and head circumference were respectively 3610"

-- "They are used to suggest various positions when a mechanical dystocia occurs and to help patients adopting them" should be "They are used to suggesting various positions when a mechanical dystocia occurs and to helping patients to adopt them."

DISCUSSION:

-- "there were no difference in participants’ " should be either "there were no differences in participants'" or "there was no difference ..."

-- "This study was not design" should be "This study was not designed"

-- "A limitation of this study consists in the absence of a reference value in the literature to differentiate between a position judged satisfactory or unsatisfactory" should be "A limitation of this study includes the absence of a reference value ..."

-- "this could inform about the efficiency" should be "this could inform about the efficacy" (or effectiveness)

-- "the role pain relief" should be "the role of pain relief" and I am not sure from the rest of that sentence exactly what you are saying, but could it be "the role of pain relief and its effect on the sense of self-empowerment."

Reviewer #4: Since it is stated that future biomechanical studies will be needed, the usage of computational biomechanics is already a reality, which should be recognized in the present paper, for example with the following reference

Borges, M., Moura, R., Oliveira, D., Parente, M., Mascarenhas, T., & Natal, R. (2021). Effect of the birthing position on its evolution from a biomechanical point of view. Computer Methods and Programs in Biomedicine, 200doi:10.1016/j.cmpb.2020.105921

The description given on the text for the position used on the study is too vague. Improve the following sentence:

“Recent research in experimental settings, i.e. not during labor, suggested a particular position, similar to squatting while lying on the back (hyperflexion of the thighs and loss of the lumbar lordosis) (cf. Figure 1), could optimize delivery [3–5]”

No information is given in relation to the women, in relation to their body mass index, height, weight, etc. This information is relevant.

6. PLOS authors have the option to publish the peer review history of their article (what does this mean?). If published, this will include your full peer review and any attached files.

Reviewer #1: No

Reviewer #2: No

Reviewer #3: No

Reviewer #4: No

---

## [Author Response · Author response to Decision Letter 0]

15 Jun 2021

Editorial office of Plos one

5th June 2021

Ref.: “ Assessing feasibility and maternal acceptability of a biomechanically-optimized supine birth position : A pilot study” 

Dear Editorial Board Member, 

We would like to thank the reviewers for the assessment our manuscript which will improve the scientific quality of our study. Please find attached the revised manuscript, with changes highlighted using the tracked changes mode in Word. We considered each suggestion from the reviewers, and you may find below our responses to their comments.

The format of the manuscript was modified according to the guidelines 

We reviewed all of our references. To our knowledge no article were retracted. 

The data underlying the findings cannot be made freely available because of ethical and legal restrictions. An important number of variables included in this study that put together could be used to re-identify the participants. Therefore, the Swiss Association of Research Ethics Committees strictly forbids making such data freely available. However, they can be obtained upon request. Readers may contact: david.desseauve@chuv.ch to request the data.

 The data cannot be freely available for the reasons mentioned in the above statement. Data access can be only possible after scientific assessment and data

sharing agreement, detailing the type of data requested. 

We have homogenised the title in the both documents

5. Please include captions for your Supporting Information files at the end of your manuscript, and update any in-text citations to match accordingly. Please see our Supporting Information guidelines for more information: http://journals.plos.org/plosone/s/supporting-information

We did not provide any Supporting Information file

6. We note that Figure 2 includes an image of a participant in the study. 

As per the PLOS ONE policy (http://journals.plos.org/plosone/s/submission-guidelines#loc-human-subjects-research) on papers that include identifying, or potentially identifying, information, the individual(s) or parent(s)/guardian(s) must be informed of the terms of the PLOS open-access (CC-BY) license and provide specific permission for publication of these details under the terms of this license.

Please download the Consent Form for Publication in a PLOS Journal (http://journals.plos.org/plosone/s/file?id=8ce6/plos-consent-form-english.pdf). The signed consent form should not be submitted with the manuscript, but should be securely filed in the individual's case notes.

Please amend the methods section and ethics statement of the manuscript to explicitly state that the patient/participant has provided consent for publication: “The individual in this manuscript has given written informed consent (as outlined in PLOS consent form) to publish these case details”.

The participant in the photograph signed the consent form. We amended the methods section and ethics statement of the manuscript on line. 

Reviewer comments to the author

Reviewer #1: This manuscript reports observed results from a pilot study investigating feasibility and acceptability of a biomechanically-optimized supine birth position. I have below questions and comments for statistical analysis.

The description is not clear for power calculation. Please provide detail for the hypotheses. What will you compare to 80%?

We clarified our calculation for power (line 469-471). 

For qualitative variable, why was a multilevel mixed-effects linear regression used? If the qualitative variable is categorical, frequency table and Chi-squared test or Fisher’s exact test may be used.

We used this statistical model as data was not distributed normally. 

Reviewer #2: Thanks for giving me such a valuable chance to review this paper. This paper addressed an important issue in maternity care and is very well-written. The authors assessed the feasibility and maternal acceptability of a biomechanically-optimized supine birth position. Similar studies on this topic are limited; Thus this study has academic significance. On the other hand, This study also provides valuable insights into dealing with obstructed labor.

On the whole, I only have a few suggestions for revision. Comments are as following. I would be delighted to recommend this paper for publication after a minor revision. 

Background:

#1 What are“eminence-based” positions? Any examples?

More information was provided (line 127-129)

#2 “ there is growing evidence recognizing that such management can improve the alignement between the birth canal (formed by pelvic bones, lumbar spine and soft pelvis tissues) and the fetus.”

Please consider adding references to support this statement.

The references were added (line 131).

#3As is stated in the Background section, “a particular position, similar to squatting while lying on the back” could optimize delivery.

How does this position optimize delivery? In what manner? By improving the alignement between the birth canal? I believe a brief explanation may add values.

This was clarified as requested (line 129-141).

Material and method

#4 Who performed the interview two days after delivery? Any rationale for choosing this date to conduct the interview?

“standardized question was ask to all the participants during an interview that took place during the two days following the delivery.

These questions were answered (line 399-404).

Discussion

#5 It would be better to add a reference for the numbers

“According to the Swiss Federal Statistic Office, rates of emergency cesarean section and instrumental deliveries in Switzerland reached respectively 15.8 % and 11.1 % in 2017” 

The reference was included as suggested (line 755).

Reviewer #3: General Notes:

-- The position is not McRobert position -- it is McRoberts' position (or McRoberts' maneuver). It was named for Dr. William McRoberts. It can be written without the possessive apostrophe, if desired -- but the s is required.

The correction was made throughout the manuscript.

-- Information on how this position was maintained (e.g. use of foot rests) would be helpful (referring to Figure). Those who understand McRoberts' position normally expect it to be maintained by two assistants/family members (one on each leg).

Mention of foot rests was included in the study population section (line 336).

-- While you refer to your papers on an optimal birth position, it would be very helpful to provide a little more detail in the Introduction of this paper about the findings in those papers. What is the necessary angle of the pelvis? How did you define optimized? How is it different in angles from lithotomy? 

This was added to the manuscript (line 129-141) 

Substantive Corrections or Clarifications:

INTRODUCTION:

-- I am not sure what you mean by "the pelvic inlet plan" -- please clarify

The sentence was clarified (line 132-134)

RESULTS:

-- what is "important back discomfort"? 

This part was replaced by “low back pain related to labor and increased in any supine position” to increase precision (line 576-577).

-- it appears from the Results section that when you discussed peripheral nerve disfunction, that you were thinking about maternal nerves. This should be clarified in the Methods section (outcomes paragraph). Readers familiar with the fetal risk factors associated with labor dystocia and the typical use of McRoberts' maneuver in shoulder dystocia events may assume that the description refers to neonatal nerve injuries.

This was clarified by adding the adjective “maternal” before “peripheral nerve dysfunction” (line 461).

-- I would suggest putting a percent sign in the parenthetical values of Table 1 to remind readers what you are signifying, as it is a bit hard to ascertain when first looking at the table.

The modification was made as suggested for all the tables.

-- I would suggest separating out the 3 time points for Apgar scores onto three separate lines in Table 2 and Table 3. This will clarify it for readers.

The modification was made as well.

DISCUSSION:

-- in your paragraph discussing the limitation due to the lack of a literature value on position, I originally thought that you were talking about the McRoberts' position. You should clarify in this sentence that you are discussing the subjects' opinions regarding their satisfaction with the position.

This was clarified as requested (line 760-761)

-- You say that you are comparing your results to randomized trials related to effectiveness of various positions (refs 14-17). However, you have not reported on any data regarding effectiveness -- only patient satisfaction. If there are specific data that you are comparing to (e.g. patient satisfaction with the positions used in those studies), please cite that data and clearly describe the comparison.

This paragraph was modified to make it more clear as we attended only to draw a comparison on patient’s satisfaction (line 760-767). 

-- "while further studies are necessary to precise the numbers" is not grammatically correct - and I am not sure exactly what you are trying to say. Increasing the numbers would allow you to statistically differentiate between narrower groups -- but it is not likely to increase the precision of your measurements. Please write this section more clearly to differentiate between actual measurements and statistical conclusions.

This was clarified in our manuscript (767-769). 

Grammatical Corrections (NOTE: There are additional grammar errors that I did not note. Please have the document reviewed by a proficient writer whose first language is English to minimize errors.)

A native speaker amended grammatical error throughout the manuscript

ABSTRACT:

-- it is most common to refer to the second stage as "second stage OF labor" rather than without the preposition

-- "is similar to the McRobert maneuver" should be "is similar to the McRoberts' maneuver"

-- In "A sub-group analyses was performed according to the median of the patient satisfaction score, to assess eventual differences between more and less satisfied patients," analyses should be ANALYSIS and the comma should be deleted between "score" and "to"

All these modifications were made in the text.

INTRODUCTION:

-- "threshold largely exceeded in developed countries" should be preceded by "a" as in "a threshold largely exceded..."

-- change "approach of childbirth" to "approach to childbirth"

-- delete "in future researches." If you really want to keep it (it doesn't make sense), it should be "in future research."

-- "can be transfer to" should be "can be transferred to"

-- "could make parturient feel" should be "could make a parturient feel"

-- "resembling at the posture" should be "resembling the posture"

We modified these grammatical errors in the manuscript.

MATERIALS AND METHODS:

-- "prelabor rupture of membrane" should be "prelabor rupture of membranes"

-- "whose fetus did not engage" should be "those whose fetus ..."

-- in the discussion of the IRB approval, there should be no comma between the IRB protocol number and "and" [i.e. -00872) and declared] and there should be no period at the end of ClinicalTrials.gov

-- "would allow to appreciate the effectiveness of the intervention without being to constraining for the women" should be "would all us to appreciate ... without being too constraining ..."

-- "the primary outcomes" should be "the primary outcome"

-- if you want to include the comma in the first sentence about outcomes, then you need a subject in the second phrase (i.e. "and this was evaluated.")

-- "to evaluate participant's satisfaction" should be either "to evaluate a participant's satisfaction" or "to evaluate participants' satisfaction"

-- "A standardized question was ask to all" should be "A standardized question was asked of all"

-- "obstetrical conditions and obstetrical outcomes between women the women with higher satisfaction to the women with lower satisfaction" should be "obstetrical conditions and obstetrical outcomes between the women with higher satisfaction and the women with lower satisfaction."

-- "Data concerning risks factor of non-engagement" should be "Data concerning risk factors of non-engagement" and "peripheral nerves disorders" should be "peripheral nerve disorders"

-- "Shapiro-wilk test" should be "Shapiro-Wilk test" as it refers to a name (or pair of names)

All these modifications were made in the manuscript.

RESULTS:

-- "median weight and head circumference were respectively of 3610" should be "median weight and head circumference were respectively 3610"

-- "They are used to suggest various positions when a mechanical dystocia occurs and to help patients adopting them" should be "They are used to suggesting various positions when a mechanical dystocia occurs and to helping patients to adopt them."

We modified these grammatical errors in the manuscript.

DISCUSSION:

-- "there were no difference in participants’ " should be either "there were no differences in participants'" or "there was no difference ..."

-- "This study was not design" should be "This study was not designed"

-- "A limitation of this study consists in the absence of a reference value in the literature to differentiate between a position judged satisfactory or unsatisfactory" should be "A limitation of this study includes the absence of a reference value ..."

-- "this could inform about the efficiency" should be "this could inform about the efficacy" (or effectiveness)

-- "the role pain relief" should be "the role of pain relief" and I am not sure from the rest of that sentence exactly what you are saying, but could it be "the role of pain relief and its effect on the sense of self-empowerment."

We modified these grammatical errors in the manuscript.

Reviewer #4: Since it is stated that future biomechanical studies will be needed, the usage of computational biomechanics is already a reality, which should be recognized in the present paper, for example with the following reference

Borges, M., Moura, R., Oliveira, D., Parente, M., Mascarenhas, T., & Natal, R. (2021). Effect of the birthing position on its evolution from a biomechanical point of view. Computer Methods and Programs in Biomedicine, 200doi:10.1016/j.cmpb.2020.105921

We appreciated your suggestion and added the corresponding reference (line 878).

The description given on the text for the position used on the study is too vague. Improve the following sentence:

“Recent research in experimental settings, i.e. not during labor, suggested a particular position, similar to squatting while lying on the back (hyperflexion of the thighs and loss of the lumbar lordosis) (cf. Figure 1), could optimize delivery [3–5]”

The sentence was improved (line 129-141)

No information is given in relation to the women, in relation to their body mass index, height, weight, etc. This information is relevant. 

We added the information concerning patients’ body mass, weight and height. There was no significant difference between the two groups. Result can be found in table 1 and 3. 

David Desseauve

---

## [Editor Report · Decision Letter 1]

31 Aug 2021

Assessing feasibility and maternal acceptability of a biomechanically-optimized supine birth position : A pilot study

PONE-D-20-31050R1

Dear Dr. Desseauve,

We’re pleased to inform you that your manuscript has been judged scientifically suitable for publication and will be formally accepted for publication once it meets all outstanding technical requirements.  In support of full transparency, I participated as a reviewer for the initial evaluation of this manuscript.

Kind regards,

Michele J. Grimm, Ph.D

Guest Editor

PLOS ONE

Additional Editor Comments (optional):

The paper is very much improved, thank you. There are a few minor corrections that I would encourage you to make prior to final publication, listed below. Also, please make certain that the data in the tables are aligned with the correct description lines in the page proofs.

Lines 97-99 -- although you include Refs 3-5 in the prior sentence, it is not clear from reading this paragraph that the "recent research" that is discussed in those references is the same as the "in theory" work discussed previously. I would suggest that you repeat the citation of Refs 3-5 at the end of the sentence (Line 99)

Line 174 -- delete "the" before one of the authors

Line 177 -- while you have provided a direct translation from the French for the question posed to the subjects, it is very awkward in English. I suggest using one of the following:

(a) "Could you report your satisfaction about the optimized position that you were requested to use on this visual scale."(b) "Could you report your satisfaction about the optimized position that was proposed to you on this visual scale."

Line 199 - "concerns" would more appropriately be "affects"

Line 240 - I think that you mean "low back pain related to labor THAT increased in any supine position." (capitalization used for clarity)

Table 3 - Maternal weight does not have a p-value presented, was this an oversight?

Table 3 - The obstetrical outcomes categories only have a single p-value, implying that the 3 or 4 parameters were assessed for overall difference between the groups (similar to an ANOVA). If so, make certain that the p-value is displayed in the page proofs so that it is clear that it refers to all categories of each parameter.

Line 344 - add "the" before pelvic inlet

Line 348 - As it is a new paragraph, it is not clear what "this" refers to. Do you mean "this study"? or "biomechanical analysis"

Line 356 - "fundamental" would more appropriately be "important"

Line 362 - "considered as" would more appropriately be "considered to be"

---

## [Editor Report · Acceptance letter]

3 Sep 2021

PONE-D-20-31050R1 

Assessing feasibility and maternal acceptability of a biomechanically-optimized supine birth position : A pilot study 

Dear Dr. Desseauve:

I'm pleased to inform you that your manuscript has been deemed suitable for publication in PLOS ONE. Congratulations! Your manuscript is now with our production department. 

Kind regards, 

on behalf of

Dr. Michele J. Grimm 

Guest Editor

PLOS ONE